# Development of the Efficient Electroporation Protocol for *Leuconostoc mesenteroides*

**DOI:** 10.3390/ijms262411933

**Published:** 2025-12-11

**Authors:** Kseniya D. Bondarenko, Leonid A. Shaposhnikov, Aleksei S. Rozanov, Alexey E. Sazonov

**Affiliations:** Scientific Center of Genetics and Life Sciences, Sirius University of Science and Technology, 354340 Sirius, Russia

**Keywords:** lactic acid bacteria, *Leuconostoc mesenteroides*, genetic transformation, electroporation, plasmid vectors, restriction–modification systems, transformation efficiency

## Abstract

*Leuconostoc mesenteroides* is a key microorganism in food biotechnology, valued for its production of flavor-forming metabolites and exopolysaccharides, and its inclusion in starter cultures and biocatalytic systems. However, the application of advanced genetic tools to *L. mesenteroides* remains hindered by multiple barriers, including inefficient DNA transfer, elevated endogenous nuclease activity, and restriction–modification systems sensitive to plasmid methylation patterns. As a result, even widely accepted electroporation methodologies often yield inconsistent or irreproducible transformation results, limiting the strain’s amenability to metabolic engineering and synthetic biology applications. In this study, a reproducible electroporation protocol for the *L. mesenteroides* strain H32-02 Ksu is developed and experimentally validated. The protocol concept relies on the sequential optimization of key process steps: targeted weakening of the cell wall followed by osmotic protection, the development of a gentle electrical stimulus that ensures membrane permeability without critical damage, and the creation of recovery conditions that minimize loss of viability and degradation of incoming DNA. Matching plasmid methylation to the recipient’s restriction profile proved critical: choosing a source for plasmid DNA production with a compatible methylation pattern dramatically increased the likelihood of successful transformation. In our case, the selection of an *E. coli* strain with a more suitable methylation profile increased the yield of transformants by 3.5 times. It was also shown that reducing the pulse voltage increase transformant number by 3 times. The combined optimization resulted in an approximately 40-fold increase in transformation efficiency compared to the baseline level and, for the first time, provided consistently reproducible access to transformants of this strain. The highest transformation efficiency was achieved: 8 × 10^2^ CFU µg^−1^ DNA. The presented approach highlights the strain-specificity of barriers in *Leuconostoc* and forms a technological basis for constructing strains with desired properties, expressing heterologous enzymes, and subsequently scaling up bioprocesses in food and related industries. The methodological principles embodied in the protocol are potentially transferable to other lactic acid bacteria with similar limitations.

## 1. Introduction

Lactic acid bacteria (LAB) are a phylogenetically diverse group of microorganisms widely used in the food industry, agriculture, medicine, and biotechnology. Due to their long history of safe use in human nutrition, many LAB have received Generally Recognized as Safe (GRAS) status for consumption [1]. This opens up wide possibilities for their industrial use, for the creation of new products and biotechnological processes, including those related to food technology [2]. Lactic acid bacteria are in demand in various industries due to their ability to produce lactic acid and synthesize beneficial metabolites [3]. LAB are used for traditional food fermentation and for the creation of functional and probiotic supplements. They are also used to produce a variety of bioproducts, as they synthesize a wide range of biologically active compounds: γ-aminobutyric acid (GABA), exopolysaccharides (EPS), conjugated linoleic acid (CLA), bacteriocins, reuterin, and reutericyclin [2]. In addition, they produce specific enzymes necessary for extracting bioactive substances from raw materials—for example, enzymes for the release of polyphenols, bioactive peptides, and the formation of inulin-type fructans and β-glucans. These metabolites have numerous beneficial properties: they improve mineral absorption, protect against oxidative stress, lower blood glucose and cholesterol levels, prevent gastrointestinal infections, and improve cardiovascular health [2]. Due to these characteristics, LAB are considered universal cell factories for nutrition and health biotechnology. Modern strains used in high-tech processes in most cases require modification using directed evolution and genetic engineering. Modern genetic technologies, including genetic engineering and genome editing (e.g., CRISPR-Cas), are already being actively applied to LAB, allowing for targeted improvement of their properties and functionality [4]. While genetically modified LAB raise certain public concerns and are subject to special regulations, the introduction of safe selection systems (e.g., food markers) and biocontainment measures provide pathways for their responsible use [4]. Tools for modifying the LAB genome have now been developed, in particular tools based on the CRISPR/Cas systems, λRed [5], as well as shuttle vectors for gene expression in ICD, including food (marker-free) systems and vectors for integration into the chromosome, which allows for fine-tuning of bacterial metabolism without introducing unwanted genetic sequences [6].

Despite the enormous biotechnological potential, genetic modification of lactic acid bacteria remains a complex task. One of the main limitations is the extremely low transformation efficiency: in laboratory strains, it typically does not exceed 10^4^–10^6^ CFU µg^−1^ DNA, while in many industrial or wild strains, it barely reaches 10^2^ or does not occur at all [6,7]. Furthermore, protocols developed for one species or strain are often inapplicable to others due to the specific characteristics of each. As a result, individual optimization of transformation conditions for each strain is required. Electroporation is the primary method for introducing foreign DNA into Gram-positive bacteria, as it is the simplest and most readily available method in modern laboratories. Exposure to an electrical pulse causes the formation of pores in the cell membrane, facilitating the penetration of plasmid DNA into the cell. However, the efficiency of electroporation is extremely sensitive to a variety of parameters: important factors include the physiological characteristics of the strain (including the presence of restriction-modification systems), the properties of the introduced plasmid (e.g., DNA methylation pattern), the parameters of the electrical pulse (field strength, duration, resistance), the use of agents to weaken the cell wall, the composition of buffers for washing cells, electroporation media, and post-pulse recovery [6,7]. Modern approaches to increasing the level of LAB transformation include preliminary weakening of the cell wall with chemical agents (e.g., adding glycine, salts, lysozyme, and penicillin during culture growth) and optimizing physiological conditions. Thus, it has been established that the addition of glycine can significantly increase the number of transformants in different LAB: for example, for *Lactobacillus plantarum,* increasing the glycine concentration in the medium to 6% led to a 30- to 100-fold increase in transformation efficiency, while for some *L. casei*, only 0.5–1% glycine proved optimal [8]. In other cases, high osmotic pressure (addition of ~0.9 M NaCl) gave a comparable effect of softening the wall and increasing the number of transformant colonies to ~10^5^ CFU µg^−1^ in *L. casei* [9]. According to the data presented in the review [6], the concentrations of such additives must be individually selected for each strain. When using high concentrations of glycine or salts, osmoprotectants (most often 0.5–1 M sucrose) must be added to the medium in the cell wash and recovery buffers to prevent osmotic lysis [10,11]. An important factor in transformation efficiency is the selection of the bacterial culture growth phase (usually with an optical density at 600 nm in the range of 0.3–0.8), at which the cells are most competent to capture DNA. In the last decade, alternative methods for delivering DNA to LAB have been developed—conjugation, phage transduction, and biolistic transformation—but electroporation remains the simplest and most versatile method, which can be relatively quickly adapted to a new strain under the right conditions [6].

In our study, we investigated the transformation capacity of a strain of *Leuconostoc mesenteroides*, a facultative anaerobic heterofermentative lactic acid bacterium widespread in nature and playing an important role in food fermentations [12,13]. *L. mesenteroides* has coccoid cells measuring ~0.5–1 µm, forms small grayish colonies <1 mm in diameter, and is known for its unique biocatalytic properties with respect to carbohydrates. During fermentation of glucose and other hexoses, *L. mesenteroides* produces equimolar amounts of D-lactate, ethanol, and CO_2_ via a combination of the hexose monophosphate and pentose phosphate pathways [13,14,15]; it is also capable of converting citrate to diacetyl and acetoin [16], and sucrose to dextrans and levans [17,18,19,20]. Dextran production is one of the key industrial applications of *L. mesenteroides*: dextran polysaccharides are used as plasma substitutes, anticoagulants (in place of heparin), and in cosmetics [21,22,23,24]. Dextran gels (e.g., Sephadex) obtained using *L. mesenteroides* are particularly valuable for protein separation (strain NRRL B-512F produces commercial dextran with high yields and a predominance of α-1,6-linkages) [17,25].

Recent studies have revealed the significant probiotic potential of *L. mesenteroides*. Some strains can exert beneficial effects in animal models. For example, the *L. mesenteroides* EH-1 strain, isolated from Mongolian cottage cheese, produces butyric acid and, when administered as a live culture, reduces blood glucose levels and the proinflammatory cytokine IL-6 in mice with type 1 diabetes [26]. Other isolates, such as strains B7 and Z8 from raw camel milk in Algeria, are characterized by pronounced antagonistic activity against pathogens: they significantly inhibit the growth of *Listeria* spp. due to the synthesis of bacteriocin (identified as leucocin B) and simultaneously show high probiotic potential in vitro—they survive at very low pH (2–3) and in the presence of 0.5–2% bile salts [27]. These strains are considered promising as new probiotic starters, combining technologically valuable properties with antimicrobial activity. Strains of *L. mesenteroides* that are capable of preventing the formation of biofilms of pathogenic bacteria by releasing antimicrobial metabolites have also been identified. In particular, the metabolic products of *L. mesenteroides* strain J.27 (isolated from kimchi) effectively inhibit the formation of biofilms of pathogens such as *Vibrio parahaemolyticus*, *Pseudomonas aeruginosa*, and *E. coli* on food substrates and food equipment surfaces [28]. *L. mesenteroides* is included in the list of safe microorganisms approved for use in food technology. Its metabolites can be used as natural additives to improve the structure, flavor, and aroma of foods (for example, polyfructose extract for dairy and bakery products), and the bacteria themselves can be used as starters in the production of fermented products (dairy, vegetables, etc.).

Despite decades of electroporation development, many *Leuconostoc* strains remain effectively non-transformable due to strong endogenous restriction systems. Therefore, establishing a reproducible protocol for DNA introduction into *L. mesenteroides* represents a necessary technological foundation enabling future molecular studies, genome engineering, and applied biotechnology. Members of the genus *Leuconostoc* have long been considered among the most difficult LAB to transform. It was only in the late 1980s that the first successful results were obtained: for example, the feasibility of electroporative plasmid insertion into *L. paramesenteroides* was demonstrated [29,30]. The researchers then adapted electroporation conditions for several food-grade *Leuconostoc strains,* including *L. mesenteroides* subsp. *cremoris*, *L. mesenteroides* subsp. *dextranicum*, and additionally *L. lactis* [31]. However, the transformation efficiency remained extremely low, and many strains did not produce transformants at all.

The main obstacles to introducing exogenous DNA into *Leuconostoc* cells are their robust cell wall and restriction-modification systems that destroy foreign DNA. Various approaches are used to increase cell wall permeability before electroporation. One of the most effective is the addition of substances to the growth medium that disrupt peptide synthesis; in the case of LAB, this is usually glycine. In addition to chemical treatment, enzymatic treatment is also used: for example, researchers achieved a 100-fold increase in the number of transformants in *L. mesenteroides* ATCC 8293 by sequentially treating the cells with a subinhibitory concentration of penicillin and 600 U mL^−1^ lysozyme before electroporation [32]. This protocol has also been successfully applied to other species of the genus *Leuconostoc* (*L. gelidum*, *L. fallax*, *L. argentinum*), as well as to several other LAB genera [32]. Of particular interest is the natural competence of some *Leuconostoc* species: for example, *L. carnosum* 4010 is capable of spontaneously taking up plasmid DNA (without electroporation) at a frequency of ~10^−6^ transformants per cell [33]. However, the practical use of natural transformation in this area is still limited, so electroporation remains the primary and most accessible method of DNA delivery.

The development of reliable transformation strategies for *Leuconostoc* is essential because recombinant LAB are widely used to enhance food fermentation processes, overproduce bioactive metabolites, and deliver therapeutic molecules as GRAS platforms. Genetically engineered LAB strains have already demonstrated improvements in metabolite yields and reduction of undesirable fermentation by-products. However, progress in this direction is limited by the absence of robust and reproducible DNA delivery systems for many LAB species, and *Leuconostoc mesenteroides* in particular. Electroporation-based transformation protocols have been reported for several LAB species and for a limited number of *Leuconostoc* strains. However, these procedures were typically developed for different *Leuconostoc* subspecies or for laboratory strains and cannot be directly applied to industrial isolates such as H32-02 Ksu. In preliminary experiments, applying published LAB electroporation recipes to H32-02 Ksu resulted in either no transformants or very low, poorly reproducible efficiencies. This motivated a systematic, strain-tailored optimization of cell-wall weakening, washing and recovery conditions, pulse parameters and DNA methylation status to establish a practical protocol for this particular production strain.

Expanding the arsenal of genetic methods for *L. mesenteroides* opens up new opportunities for biotechnological modernization. A robust transformation system enables the creation of recombinant strains with improved characteristics and new functions. For example, this makes it possible to construct phage-resistant and stress-resistant production crops, selectively modify metabolic pathways to enhance the production of target metabolites (flavor compounds, texture-forming polymers, etc.), and create probiotic “chassis” strains for the delivery of therapeutically important molecules. A practical example of this approach is the reduction of D-lactic acid byproduct formation during sauerkraut fermentation using a recombinant *L. mesenteroides* strain expressing the L-lactate dehydrogenase gene [34]. Thus, the development of *L. mesenteroides* transformation methods lays the foundation for the full realization of its biotechnological potential in various industries.

## 2. Results and Discussion

### 2.1. Analysis of Factors Influencing Transformation

Based on literature data and our experience, successful electroporation of new lactic acid bacteria strains requires comprehensive optimization of several key parameters. We identified the following limiting factors that require attention when developing the protocol:Cell wall condition: selection of nutrient medium composition and wall-loosening agents when culturing cells for biomass production. Weakening of the peptide glycan (e.g., with glycine, high sodium chloride concentrations, or enzymatically) significantly increases cell permeability [8,9,11].Cell growth phase: determining the optimal optical density of the culture for electroporation. Typically, cells in mid-logarithmic growth (OD_600_ ~0.4–0.6) yield the highest yield of transformants, while cells in the stationary phase are almost non-transformable [11,35].Buffer solutions: selecting an appropriate wash buffer and recovery medium after electroporation. Osmotically supporting sucrose-based buffers (0.5–1 M) are typically used for all steps: wash, electroporation, and post-shock recovery to prevent cell lysis [8,10,11,36].Electrical pulse parameters: optimization of electroporation conditions—cuvette size (0.1 or 0.2 cm gap), voltage, resistance and capacitance in the circuit, and pulse duration. Standard conditions for LAB electroporation in the literature include the use of 0.1 cm cuvettes and pulse parameters of 2 kV, 200 Ω, 25 μF, and 5–6 ms [37].Plasmid DNA source: selecting a plasmid production system in *E. coli*, taking into account the methylation pattern. The presence of Dam/Dcm tags on DNA can lead to its cleavage by MAB restriction enzymes. Therefore, it is preferable to use plasmids isolated from Dam^−^/Dcm^−^ *E. coli* strains or from in vitro expression, which, in some cases, has been reported to increase transformation efficiency by 3–4 orders of magnitude [38].Selective markers and recovery: determining the minimum inhibitory concentration of the antibiotic for a given strain and allowing sufficient cell recovery time before plating on selective media. It is essential to ensure that the strain does not exhibit natural resistance to the selected antibiotic; otherwise, selection of transformants will be impossible.

Based on literature data, a general scheme for electroporation of *L. mesenteroides* was developed, used for further optimization and presented in Figure 1.

The parameters explored in this study—glycine supplementation, washing and recovery conditions, electrical field strength and antibiotic concentration—are classical variables in electroporation optimization. The novelty of our work does not lie in identifying entirely new categories of parameters, but rather in demonstrating that a carefully tuned combination of these factors is essential to convert the industrial *L. mesenteroides* strain H32-02 Ksu from a practically non-transformable organism into one that can be manipulated reproducibly. In that sense, the contribution of this study is contextual and enabling: it removes a technical barrier that has so far hindered genetic and molecular analyses in this particular *Leuconostoc* strain.

Based on the literature, a primary protocol for screening strains for transformability was developed: MRS with 1% glycine was used for cell growth, 10 mM MgCl_2_ solution, then 0.5 M sucrose solution with 10% glycerol were used as wash buffers, electroporation was carried out in 0.1 cm cuvettes at a voltage of 2.0 kV, a resistance of 200 Ω, and a capacitance of 25 μF (the resistance and capacitance remained constant in all experiments, so we will omit mentioning these parameters below), MRS with 0.5 M sucrose was used as a recovery medium. The pHyC plasmid (Figure 2), isolated from the *E. coli* Top10 strain, was used as a transformation vector.

During the work, transformability screening was carried out for 2 strains of *Leuconostoc mesenteroides*: XK25-02 and H32-02 Ksu from the microbial collection of the Sirius University of Science and Technology. The *Leuconostoc mesenteroides* H32-02 Ksu was successfully transformed. Four colonies were obtained per plate, and the presence of the pHyC vector was assessed using colony PCR (Figure 3).

Based on the results obtained at this stage, the *L. mesenteroides* H 32-02 Ksu strain was selected for further optimization of the electroporation protocol.

### 2.2. Optimization of Growth Conditions with Glycine

Three concentrations of glycine were used: 1%, 2%, or 2.5%. In all cases, the cells were grown in 15 mL of MRS + glycine for 2 days, and then additionally in 35 mL of fresh MRS + glycine to an OD_600_ of ~0.5, after which standard washing and electroporation with the same amount (200 ng) of DNA were performed. When culturing in the presence of 2% and 2.5% glycine, transformation did not occur. In the presence of 1% glycine, the transformation efficiency was 20 ± 5 CFU µg^−1^ DNA. Thus, the use of higher glycine concentrations (2% or 2.5%) had an inhibitory effect on the competence of the cells during transformation. It is important to note that, according to the literature data for other species, the optimal dose of glycine varies. E.g., for *L. casei*, an effective glycine concentration in the range of 0.5–1% is reported [11], while for individual strains of *L. plantarum* the optimal concentration reaches 6–8% [11]. It is likely that excessive weakening of the *L. mesenteroides* cell wall hinders plasmid penetration due to uneven pores. This emphasizes the need to tailor the glycine concentration to each strain individually.

### 2.3. The Influence of Pulse Parameters on Transformation Efficiency and Culture Survival

To study the effect of electrical pulse voltage during electroporation, cuvettes with a gap of 0.1 cm and 0.2 cm were used, and the voltage was varied while other parameters were kept constant (resistance 200 Ω, capacitance 25 μF, pulse duration 5–6 ms). The voltage was chosen to be 1.7 kV, 2 kV, 2.5 kV, and 3 kV. Using cuvettes with 0.2 cm electrode gap significantly reduced the transformation efficiency (down to 0), all other conditions being equal. Using voltages of 2.5 kV and 3 kV proved lethal for cells both when using 0.1 cm and 0.2 cm cuvettes. At a voltage of 2 kV in 0.1 cm cuvettes, the transformation efficiency averaged 15 colonies or 75 ± 3 CFU µg^−1^ DNA. When using cuvettes with a 0.2 cm gap, the efficiency averaged 20 ± 5 CFU µg^−1^ DNA. The highest transformation efficiency was observed at a voltage of 1.7 kV, averaging 225 ± 8 CFU µg^−1^ DNA when using cuvettes with a 0.1 cm gap between the electrodes and 45 ± 6 CFU µg^−1^ DNA for 0.2 cm cuvettes. The results are presented in Figure 4.

### 2.4. The Effect of Post-Impulse Recovery Medium

Three variants of recovery media were tested: MRS, MRS supplemented with 0.5 M sucrose, and MRS with 0.5 M sucrose and 0.1 M MgCl_2_. The standard MRS medium was insufficient for cell survival—without sucrose, most cells died, and even on plates without antibiotics, no growth occurred. In the other two variants, adding sucrose to the recovery medium significantly increased cell survival—clearly visible colonies grew on dishes with antibiotics, while a continuous lawn was observed on dishes without antibiotics. However, adding magnesium chloride to the recovery medium did not significantly affect transformation efficiency, so MRS medium with 0.5 M sucrose was chosen for the final protocol.

Analysis of the incubation time in the recovery medium showed that 3 h of incubation after the electrical pulse was sufficient to restore cells (efficiency 225 ± 3 CFU µg^−1^ DNA). With shorter recovery times, transformant colonies either did not appear at all (1 and 2 h of incubation after recovery), or the transformation efficiency dropped significantly—with 2.5 h of recovery, the efficiency was 100 ± 5 CFU µg^−1^ DNA. Increasing the recovery time to 4 h did not significantly improve transformation efficiency (Figure 5).

### 2.5. Effect of Chloramphenicol Concentration

The concentration of the antibiotic varied from 5 μg mL^−1^ to 25 μg mL^−1^. Concentrations below 5 μg mL^−1^ resulted in non-selective growth of untransformed cells on plates, while concentrations above 25 μg mL^−1^ did not result in colony formation (Figure 6). 7 μg mL^−1^ of chloramphenicol was chosen as the optimal concentration, since using this concentration of the antibiotic, as well as 5 and 6 μg mL^−1^, resulted in maximum selective growth of transformed colonies, while using the higher of the three concentrations reduced the risk of non-selective growth on plates.

### 2.6. Effect of DNA Methylation and Wash Buffers

*E. coli* strains differ in the activity of Dam and Dcm methyltransferases, which methylate specific sites on DNA. The bacterial restriction-modification systems (particularly the LAB) are capable of destroying unfamiliar (differently methylated) DNA [6]. It has been reported that the use of unmethylated DNA (from Dam^−^/Dcm^−^ *E. coli* strains) can increase the efficiency of LAB transformation by several orders of magnitude [6]. To test the effect of different DNA methylation status on the transformation efficiency, we replicated the pHyC plasmid in three *E. coli* strains (Ec135, Top10, MC1061) and then transformed *L. mesenteroides* H32-02 Ksu with the plasmid with three different methylation statuses. Three variants of cell washing before the electrical pulse were also tested. The following solution combinations were used for washing: (1) 10 mM MgCl_2_ for the first washing step and 1 M sucrose for the second; (2) 10 mM MgCl_2_ for the first step and 0.5 M sucrose with 10% glycerol for the second; (3) 1 M sucrose for the first and second steps. The results are presented in Table 1. The methylation systems present in the strains used were as follows: Dam methylates the N6 position of adenine in the GATC sequence, Dcm methylates the internal (second) cytosine in the CCAGG and CCTGG sequences (at the C5 position of cytosine), EcoKI methylates adenine in the AAC(N6)GTGC and GCAC(N6)GTT sequences.

The results of the experiment using various cell washing buffers show that a 1 M sucrose concentration is the most optimal. However, using a 1 M sucrose solution alone is not an option in this case; pre-washing the cells with a 10 mM MgCl_2_ solution is also crucial for the results. This is because almost all LAB synthesize extracellular polysaccharides and teichoic acids, which support the cell wall structure. Polysaccharides and teichoic acids impart a negative charge to the cell wall, significantly hindering the penetration of any DNA. Magnesium ions neutralize the charge of the cell and DNA, thereby increasing the efficiency of electroporation [39].

The pattern of plasmid methylation was found to have a significant effect on the transformation efficiency of the studied strain. Among the three tested donor *E. coli* strains, the lowest transformation level was observed for the Top10 strain, which carries the Dam^+^/Dcm^+^ methylation systems. The use of plasmid DNA isolated from the Ec135 strain (deficient in the Dam^−^/Dcm^−^ restriction-modification systems) led to an approximately threefold increase in efficiency. However, when transforming with a plasmid isolated from the MC1061 strain (characterized by the presence of both the basic Dam^+^/Dcm^+^ systems and an additional EcoKI system), the efficiency increased ~3.5-fold compared to Top10. Intuitively, one could expect that the unmethylated plasmid from Ec135 should provide the highest transformation level (expected ratio Ec135 > Top10 ≥ MC1061). However, experimental data showed a different dependence—MC1061 > Ec135 ≫ Top10.

*E. coli* Top10 strain contains both basic methylases, and the plasmid from this strain is methylated at both adenine and cytosine residues. Many lactic acid bacteria (LAB) are known to recognize such tags as foreign and eliminate methylated DNA through specific type II and IV restriction-modification systems targeting methylated sites [40,41,42]. This explains the low transformation rates for plasmids isolated from Top10 (225 ± 15 CFU µg^−1^ DNA).

Ec135 strain lacks methylases, leading to expect greater transformation efficiency. This was confirmed experimentally for *L. mesenteroides* H32-02 Ksu, where an increase in transformation efficiency to 650 ± 20 CFU µg^−1^ DNA was observed. While higher from the previous *E. coli* strain the efficiency overall is still lower than the usual 10^4^–10^5^ CFU µg^−1^ DNA of industrial LAB strains which indicates the possible presence of a specific type I restriction-modification system in *L. mesenteroides* H32-02 Ksu, which recognizes unmethylated sequences and initiates DNA degradation.

The highest transformation efficiency was achieved using a plasmid from the *E. coli* MC1061 (800 ± 30 CFU µg^−1^ DNA). This strain is recommended in the literature for producing plasmids for subsequent transformation into *B. subtilis* strains.

To better interpret the methylation-dependent effects, we sequenced the genome of *L. mesenteroides* H32-02 Ksu and analyzed it for defense systems. Using PADLOC and PADLOC-DB, we identified a single complete type I restriction–modification (R–M) locus, composed of a predicted restriction endonuclease subunit and a cognate type I methyltransferase. The latter showed 97% amino-acid identity to the characterized methyltransferase M.Lme20241II and was assigned the same predicted recognition sequence, GAAYNNNNNCTT, with N^6^-methyladenine on the first adenine (GA^mAYNNNNNCTT). No additional type II, III or IV restriction endonucleases were detected in the draft genome, indicating that this type I system is likely to represent the major sequence-specific DNA restriction barrier in strain H32-02 Ksu. Besides the type I R–M locus, PADLOC also predicted several other phage-defense systems, including PDC-S01/S16/S35-like defense islands and an AbiD-type abortive infection protein, suggesting that H32-02 Ksu harbors a broader antiviral arsenal beyond classical R–M.

The dependence of transformation efficiency on the methylation status of the incoming plasmid DNA confirms that endogenous restriction–modification systems constitute a major barrier in *L. mesenteroides* H32-02 Ksu. Publicly available genomes of L. mesenteroides typically encode several type I R–M loci and additional DNA methyltransferases, and at least one strain carries a biochemically characterized type II restriction enzyme, consistent with substantial inter-strain diversity in restriction barriers [43,44,45,46,47]. Moreover, recent defense-system surveys have identified a type IV, methylation-dependent restriction endonuclease homolog (GC874) in *L. mesenteroides*, closely related to an enterococcal enzyme that targets methylated DNA [48]. These observations support the view that *L. mesenteroides* strains possess a complex and highly strain-specific arsenal of restriction activities that can discriminate between different methylation patterns on incoming DNA. In H32-02 Ksu itself, however, our genome analysis identified only a single complete type I R–M system with a predicted GA^mAYNNNNNCTT specificity (see above) and no additional type II, III or IV restriction endonucleases, suggesting that this type I system is the dominant DNA cleavage activity in this strain.

In this context, our observation that plasmid DNA isolated from MC1061 (Dam^+^/Dcm^+^/EcoKI mK^+^) transforms *L. mesenteroides* more efficiently than plasmid prepared from a Dam^+^/Dcm^+^ strain (Top10) or from a largely unmethylated strain (Ec135; Dam^−^/Dcm^−^) is most parsimoniously interpreted as a case where at least one host restriction barrier is sensitive to the overall adenine-methylation pattern. EcoKI modifies the sequence AACNNNNNNGTGC/GCACNNNNNNGTT, which is distinct from the predicted host self-motif GAAYNNNNNCTT, so direct mimicry of the host recognition site by EcoKI is unlikely. Nevertheless, the additional N^6^-adenine methylation provided by EcoKI may alter the density and distribution of methylated adenines along the plasmid in a way that partially reduces cleavage by the H32-02 Ksu type I system or by other methylation-sensitive nucleases, whereas the Dam^+^/Dcm^+^ pattern alone is a more favorable substrate. At present this remains a working hypothesis; targeted disruption of the identified type I locus and in vitro characterization of its methylation and cleavage specificity will be required to fully resolve the underlying mechanism.

To sum it up, for *L. mesenteroides* H32-02 Ksu the priority in transformation efficiency is distributed in the following order: MC1061 > Ec135 ≫ Top10, which reflects complex interactions between the plasmid methylation pattern and the restriction–modification systems of the recipient cell. To our knowledge, this is the first demonstration that the methylation background of the donor *E. coli* strain, in particular the presence of a modification-proficient EcoKI system, has a pronounced effect on the electroporation efficiency of *L. mesenteroides*.

### 2.7. Limitations of This Study and Future Directions

This study has several limitations. First, we focused on a single *L. mesenteroides* strain (H32-02 Ksu) and one shuttle vector (pHyC). Although this strain is relevant as a difficult-to-transform dairy isolate, additional work will be required to adapt and validate the protocol for other *Leuconostoc* strains and plasmid backbones. Second, although we obtained a draft whole-genome sequence and in silico predictions of defense systems for H32-02 Ksu, we did not experimentally validate the activity and recognition specificity of the identified type I R–M system or of the other predicted defense loci (PDC-type and AbiD-like systems). Our interpretation of the methylation-dependent effects therefore remains indirect and should be regarded as a hypothesis that requires functional testing. Third, while the optimized procedure increased transformation efficiency by approximately 40-fold, the maximum level obtained (8 × 10^2^ CFU µg^−1^ DNA) is still lower than those typically reported for model lactic acid bacteria.

## 3. Materials and Methods

### 3.1. Bacterial Strains, Plasmids and Culture Conditions

*Escherichia coli* strains Ec135, Top10, and MC1061 were used to produce plasmid DNA. These three strains were selected to evaluate the effect of plasmid DNA methylation on transformation efficiency, as they represent distinct methylation backgrounds: Top10 (Dam^+^/Dcm^+^), Ec135 (Dam^−^/Dcm^−^), and MC1061 (Dam^+^/Dcm^+^/EcoKI mK^+^). *E. coli* cultures were grown aerobically at 37 °C in liquid LB broth (10 g·L^−1^ tryptone, 5 g·L^−1^ yeast extract, 10 g·L^−1^ NaCl) or on LB agar medium, with the addition of antibiotics as needed. To prepare the agar medium, Bactoagar was added to the broth to 1.2% before sterilization.

The pHyC shuttle vector (4871 bp; Gram^+^/Gram^−^; p15A origin of replication for *E. coli*, repB for Gram-positive hosts, AmpR and CmR markers) was kindly provided by O.V. Berezina from the collection of the Laboratory of Molecular Biotechnology of the National Research Center for Bioinformatics and Microbiology, and was used as a transferred plasmid.

*Leuconostoc mesenteroides* strains XK25-02 and H32-02 Ksu from the microbial collection of the Sirius University of Science and Technology were isolated from cow’s milk. Milk containing strain XK25-02 was purchased at a market in the urban settlement of Khadyzhensk, Krasnodar Krai, and milk containing strain H32-02 Ksu was purchased in the village of Malinovka, Novosibirsk District, Novosibirsk Region.

*Leuconostoc* strains were cultured in liquid MRS nutrient medium (10 g·L^−1^ tryptone, 10 g·L^−1^ meat extract, 5 g·L^−1^ yeast extract, 20 g·L^−1^ glucose, 1 g·L^−1^ Tween-80, 2 g·L^−1^ ammonium citrate, 5 g·L^−1^ sodium acetate, 0.1 g·L^−1^ magnesium sulfate, 0.05 g·L^−1^ manganese sulfate, 2 g·L^−1^ disodium phosphate) at 37 °C without stirring. Before the experiments, the LAB strains from the collection were pre-cultured several times on solid MRS medium to refresh the culture and increase viability. To prepare the agar medium, Bactoagar was added to the broth before sterilization to 1.2%.

### 3.2. E. coli Cells and Their Transformation

*E. coli* culture (strain Top10, MC1061 or Ec135) was grown at 37 °C with vigorous shaking until optical density OD_600_ = 0.4–0.6 was reached. 50 mL of this culture was cooled and centrifuged (5000 rpm, 10 min, +4 °C). The cell pellet was carefully resuspended in 50 mL of cold ITB buffer (55 mM MnCl_2_, 15 mM CaCl_2_, 250 mM KCl, 10 mM HEPES, pH ~7.0) and pelleted again by centrifugation. After removing the supernatant, the cells were resuspended in 2 mL of ITB buffer (approximately 25-fold concentration relative to the original culture) and 150 µL of DMSO was added. The resulting suspension was dispensed into sterile 150 µL tubes and frozen in liquid nitrogen. Competent cells were stored at –80 °C for no more than 1–2 months.

*E. coli* transformation was performed using the heat shock method: 5–10 ng of plasmid DNA were added to cells thawed on ice, the tubes were incubated on ice for 20–30 min, then kept at 42 °C for 60 s and quickly cooled on ice. After this, 500 µL of warm LB broth were added to the cells, the cells were incubated for ~1 h at 37 °C and plated on selective LB plates with ampicillin (concentration 100 µg mL^−1^) to select *E. coli* transformants.

### 3.3. Plasmid DNA Isolation from E. coli

pHyC plasmid DNA was isolated from 5 to 10 mL of an overnight *E. coli* culture using the Plasmid Miniprep isolation kit (Eurogen, Moscow, Russia) according to the manufacturer’s protocol. The purified plasmid concentration was measured spectrophotometrically on NanoDrop One (Thermo Fisher Scientific, Waltham, MA, USA) and adjusted to a working concentration of ~100–150 ng μL^−1^. Freshly prepared plasmid preparations were used for electroporation.

### 3.4. Whole-Genome Sequencing and Genome Assembly

Genomic DNA of *L. mesenteroides* H32-02 Ksu was extracted using the standard phenol–chloroform method. Whole-genome sequencing was performed using Illumina paired-end technology. Reads were quality-filtered and assembled de novo using Unicycler v0.4.8. The draft genome was deposited in GenBank under accession JBSROM000000000 (BioProject PRJNA1373125, BioSample SAMN53635418). Genome coverage was calculated automatically by NCBI from the submitted read set.

The assembly consisted of 113 contigs with an N50 of 43,920 bp. The genome was annotated using the NCBI Prokaryotic Genome Annotation Pipeline. Restriction–modification (R–M) systems and defense genes were predicted using various web tools: PADLOC (https://padloc.otago.ac.nz/padloc/, accessed on 3 December 2025), Restriction-Modification Finder 1.1 (https://cge.food.dtu.dk/services/Restriction-ModificationFinder/, accessed on 3 December 2025), and BLAST searches against REBASE (https://rebase.neb.com/rebase/rebase.html, accessed on 3 December 2025).

### 3.5. Preparation of Electrocompetent Lactic Acid Bacteria Cells and Electroporation

All optimization experiments were performed in three independent biological replicates. Transformation efficiency values represent mean ± standard deviation.

The original transformation protocol was taken from Sakamoto’s 2002 work [49]. This protocol was further optimized during the course of this work. The optimized transformation protocol is presented below.

Strains of *Leuconostoc* were cultured on plates with MRS agar medium. Two to three colonies from a fresh plate, no more than a week old, were then transferred to 5 mL of liquid MRS and incubated for 2 to 3 days at 37 °C without stirring.

After 2–3 days of incubation, 750 µL of the culture was transferred to a Falcon-type tube with 15 mL of fresh MRS medium containing 1% glycine and grown for another 2 days at the same temperature without stirring.

On the day of the electroporation procedure, a fresh portion of 35 mL of MRS medium with 1% glycine was inoculated with the culture from the previous cultivation stage with a calculated initial optical density of about OD_600_ = 0.2, and cultured until an optical density of OD_600_ = 0.4–0.6 was reached.

Cells from the resulting culture were pelleted by centrifugation (4000 rpm, 10 min, 4 °C). All subsequent operations were performed in chilled tubes strictly on ice.

The cell pellet was gently resuspended in 10 mL of cold 10 mM MgCl_2_ and pelleted again (4000 rpm, 5 min, +4 °C). After removing the supernatant, the cells were washed twice with a chilled solution of 1 M sucrose (the first wash in an equal volume of 35 mL, and the second—in half the volume, i.e., 17.5 mL). The final pellet was resuspended in 250 μL of 1 M sucrose. The suspension was distributed in 50 μL portions into chilled sterile tubes and stored on ice until the electroporation procedure (for a maximum of 1–2 h, avoiding heating) or at –80 °C.

For each transformation, 50 µL of competent cells were taken. ~200 ng of pHyC plasmid DNA in 2 µL of water were added to the cells. The mixture was carefully transferred to an ice-cooled electroporation cuvette with an interelectrode gap of 0.1 cm. Electroporation was performed on a Bio-Rad Gene Pulser electroporator (Bio-Rad Laboratories, Hercules, CA, USA) using the following pulse mode: voltage of 1.7 kV, resistance of 200 Ω, and capacitance of 25 µF. With these parameters, the discharge duration was ~5–6 ms. Immediately after the discharge, 950 µL of warm recovery medium (MRS broth supplemented with 0.5 M sucrose) was added. The entire suspension was then transferred to a sterile tube. The cells were incubated for 3 h at the optimal temperature without shaking to allow them to recover. Afterwards, the cells were pelleted by centrifugation (10,000 rpm, 1 min), ~900 µL of the supernatant was collected, the pellet was resuspended in the remaining volume, and then plated onto selective plates with MRS agar containing 7 µg mL^−1^ chloramphenicol. A control plate of the same cells without antibiotics was also plated in parallel to assess overall survival after electroporation. All plates were incubated at the optimal temperature for 4–5 days. Colonies were counted to calculate the transformation efficiency (CFU per 1 µg of added DNA).

### 3.6. Analysis of the Transformants

The presence of the plasmid in the transformant colony was determined using PCR. A small number of cells were removed from colonies grown on selective plates using a sterile loop and suspended in 20 µL of water. 2 µL of the mixture was then used as a PCR template. PCR was performed with primers to a specific fragment of the pHyC plasmid (Table 2) using the BioMaster HS-Taq PCR-Color ready-to-use PCR mixture (Biolabmix, Novosibirsk, Russia) according to the manufacturer’s protocol. The amplification products were analyzed by electrophoresis in a 1% agarose gel, comparing the fragment size with a DNA marker. The presence of the target PCR product in a colony confirmed that it was a true transformant carrying the plasmid.

## 4. Conclusions

The optimized protocol developed in this work substantially improved the electroporation efficiency of *L. mesenteroides* H32-02 Ksu and, importantly, enabled the reproducible recovery of transformants in a strain that initially showed almost no transformability under standard conditions. Through systematic variation in cell wall weakening, washing and recovery conditions, electrical pulse parameters, and the methylation status of the incoming plasmid DNA, we achieved an approximately 40-fold increase in transformation efficiency, reaching up to 8 × 10^2^ CFU per µg of added DNA. Although this level is still lower than those attained for model laboratory LAB strains, it is sufficient to support routine cloning and genetic manipulation in H32-02 Ksu and thus represents a practical enabling advance.

Our results also underscore the strong influence of DNA methylation patterns on transformation efficiency, consistent with the presence of methylation-sensitive restriction barriers in *L. mesenteroides*. In particular, plasmid DNA propagated in an *E. coli* strain carrying a modification-proficient EcoKI system yielded the highest number of transformants, suggesting that carefully chosen methylation backgrounds can partially alleviate host restriction.

Overall, this work highlights the pronounced strain-dependence of transformation conditions in *Leuconostoc* and demonstrates that electroporation parameters need to be tailored individually for each strain. The protocol described here provides a reproducible starting point for molecular genetic studies, metabolic engineering and development of improved starter cultures based on *L. mesenteroides* and is aligned with the goals of the Special Issue by establishing essential technological groundwork for subsequent mechanistic and functional research.

## Figures and Tables

**Figure 1 ijms-26-11933-f001:**
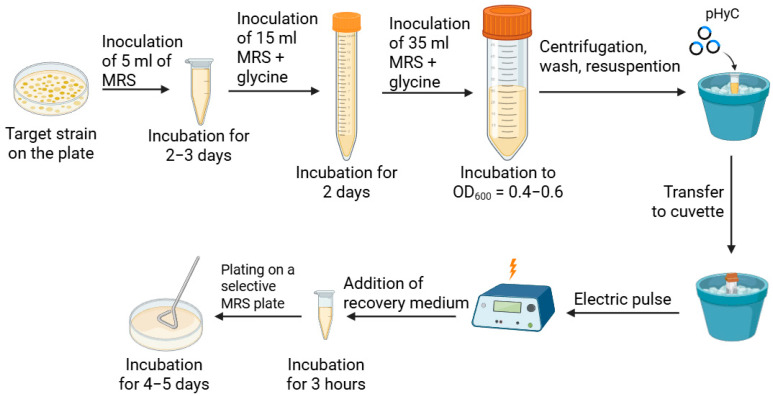
Electroporation scheme of *L. mesenteroides*, optimized in this work.

**Figure 2 ijms-26-11933-f002:**
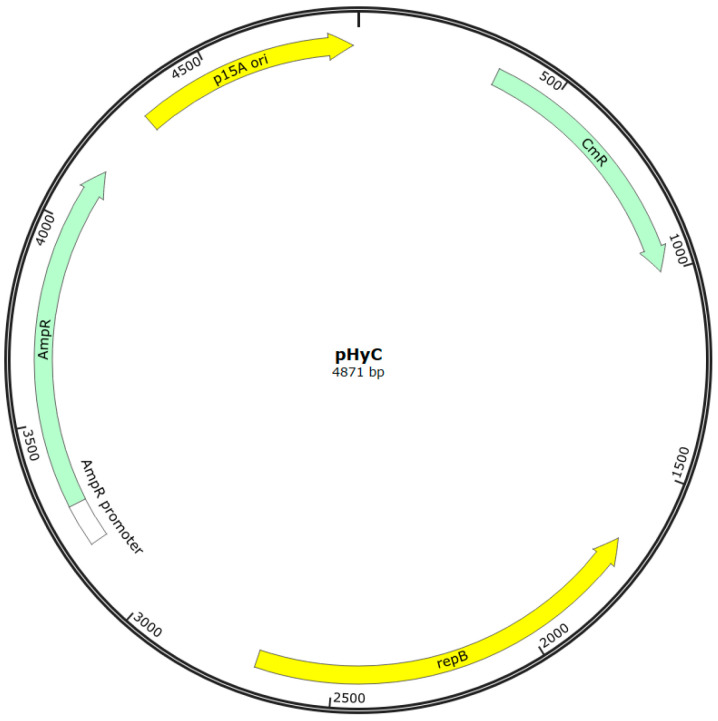
Map of the shuttle plasmid pHyC used in this study. The plasmid contains the p15A origin of replication for *E. coli* (yellow), the *repB* gene supporting replication in Gram-positive hosts (yellow), and ampicillin (AmpR, green) and chloramphenicol (CmR, green) resistance genes for selection in *E. coli* and lactic acid bacteria, respectively.

**Figure 3 ijms-26-11933-f003:**
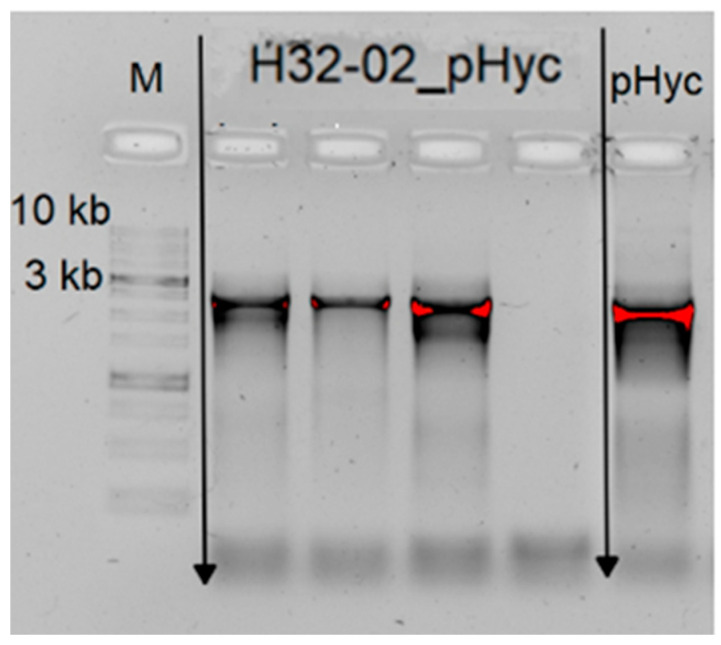
PCR confirmation of pHyC plasmid in individual transformant colonies of *L. mesenteroides* H32-02 Ksu. H32-02_pHyc: four lanes for colonies grown on chloramphenicol-containing MRS agar and analyzed by colony PCR using pHyC-specific primers described in Section 3. M: DNA molecular weight marker; pHyc: positive control PCR from plasmid DNA.

**Figure 4 ijms-26-11933-f004:**
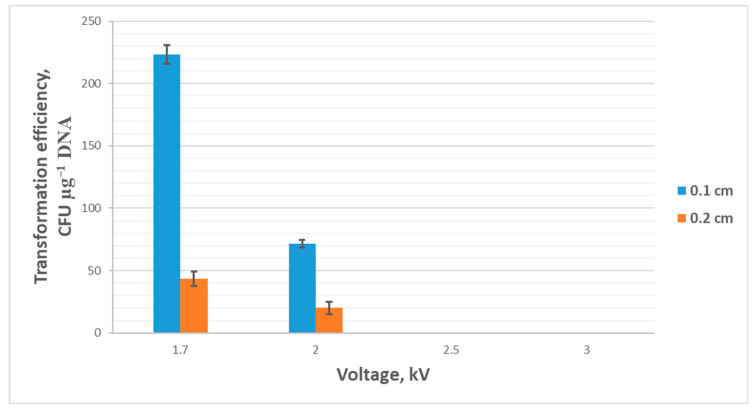
Transformation efficiency (mean ± SD, n = 3 independent experiments) dependence on pulse voltage and cuvette’s electrode gap.

**Figure 5 ijms-26-11933-f005:**
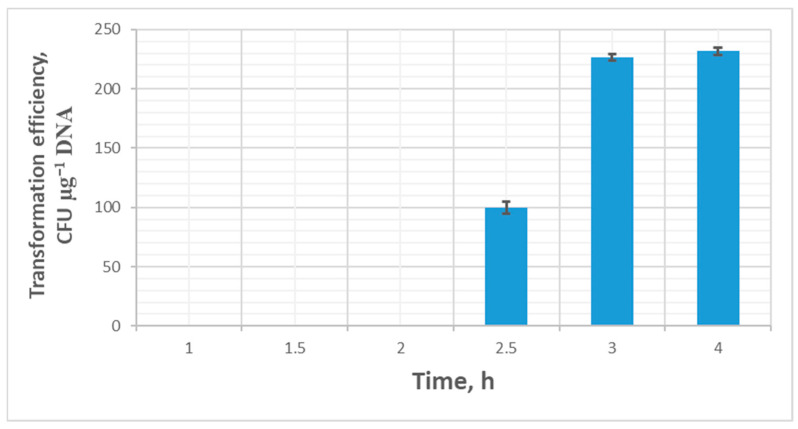
Transformation efficiency (mean ± SD, n = 3 independent experiments) dependence on the time of cell recovery after an electrical pulse.

**Figure 6 ijms-26-11933-f006:**
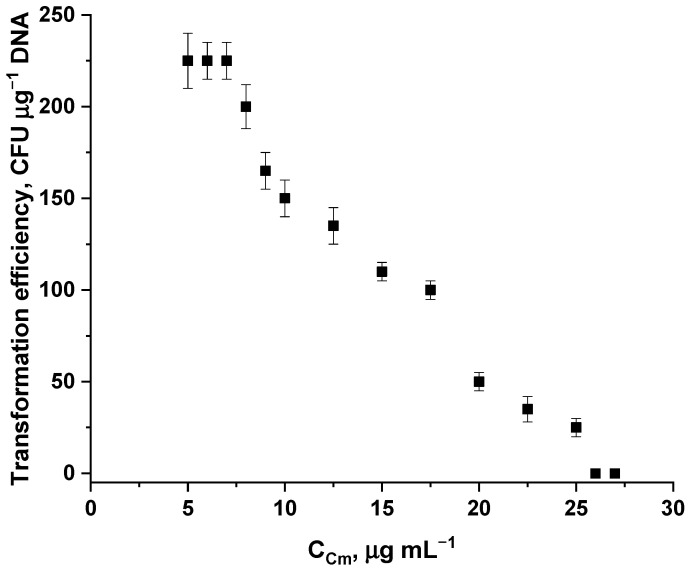
Transformation efficiency (mean ± SD, n = 3 independent experiments) dependence on the concentration of chloramphenicol in a solid nutrient medium.

**Table 1 ijms-26-11933-t001:** Transformation efficiency using different wash buffers and different *E. coli* strains for plasmid replication. Efficiency was calculated from three independent replicates for each of nine experiments. All values represent the mean ± SD of three independent experiments.

*E. coli* Strain	Methylase Activity	Wash Buffer		
		10 mM MgCl_2_ + 1 M Sucrose	10 mM MgCl_2_ + 0.5 M Sucrose with 10% Glycerol	1 M Sucrose
Top10	Dam^+^/Dcm^+^	225 ± 15	75 ± 5	0
Ec135	Dam^−^/Dcm^−^	650 ± 20	300 ± 15	0
MC1061	Dam^+^/Dcm^+^/EcoKI	800 ± 30	475 ± 30	0

**Table 2 ijms-26-11933-t002:** Specific primers for the pHyC plasmid.

Primer	Sequence 5′–3′
pHyC_f	aatctggagccggtgagcgtggaagtcgcggtatc
pHyC_r	aaaatcgctaatgttgattactttgaacttctgcatattcttg

## Data Availability

The whole-genome shotgun sequence of *Leuconostoc mesenteroides* H32-02 Ksu has been deposited in DDBJ/ENA/GenBank under the accession number JBSROM000000000 (BioProject PRJNA1373125, BioSample SAMN53635418). The version described in this paper is JBSROM010000000. All other data supporting the findings of this study are available from the corresponding author upon reasonable request.

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
