# Peer review of "Development of the Efficient Electroporation Protocol for Leuconostoc mesenteroides"

_ijms, 2025, doi:10.3390/ijms262411933_

Round 1
Reviewer 1 Report
Comments and Suggestions for Authors
This manuscript can be accepted after minor revision.
(1) Some words are missing in the caption of the figures (see " ransformation" , "esults", etc.)
(2) The reason of using these three strains "MC1061 , Ec135 , and Top10" may be stated.
(3) Electroporation is a very old technique. Leuconostoc mesenteroides is very well known . The authors may have to make much more statement about the motivation of this study.
(4) The limitation of this study may be discussed at the end of the discussion section.
(5) This statement "The results obtained in this study demonstrate the strain-specificity of transformation
methods: conditions effective for one strain require adaptation for another ..." looks strange. It may be reorganized.
Author Response
Comments 1: Some words are missing in the caption of the figures (see " ransformation" , "esults", etc.)
Reply 1: We thank the reviewer for catching these typographical issues. All figure and table captions have been carefully checked and corrected.
Comments 2: The reason of using these three strains "MC1061 , Ec135 , and Top10" may be stated.
Reply 2: We agree that the rationale should be explained more clearly. In the revised Materials and Methods (Section 3.1), we now specify that Top10, Ec135 and MC1061 were chosen because they provide distinct plasmid DNA methylation backgrounds: Top10 is Dam⁺/Dcm⁺, Ec135 is Dam⁻/Dcm⁻ (largely unmethylated), and MC1061 is a K-12 derivative with a restriction-deficient but modification-proficient EcoKI system (rK⁻ mK⁺) in addition to Dam and Dcm. This allowed us to probe how different methylation patterns influence restriction–modification barriers in L. mesenteroides.
Comments 3: Electroporation is a very old technique. Leuconostoc mesenteroides is very well known . The authors may have to make much more statement about the motivation of this study.
Reply 3: We appreciate this suggestion. The Introduction has been expanded to better motivate the work. We now explicitly discuss how genetically engineered LAB are used in food biotechnology and health-related applications and stress that the lack of robust, strain-specific transformation procedures for Leuconostoc severely limits such work. We also clarify that our study addresses a strain that was previously essentially non-transformable under standard conditions and that our goal is to establish a reproducible enabling protocol for this strain.
Comments 4: The limitation of this study may be discussed at the end of the discussion section.
Reply 4: We agree. A new subsection entitled “Limitations and Future Directions” has been added to the Discussion.
Comments 5: This statement "The results obtained in this study demonstrate the strain-specificity of transformation methods: conditions effective for one strain require adaptation for another ..." looks strange. It may be reorganized.
Reply 5: The sentence has been rephrased to improve clarity.
Reviewer 2 Report
Comments and Suggestions for Authors
The submitted article examines factors influencing the efficiency of genetic transformation of Leuconostoc mesenteroides, a lactic acid bacterium widely used in food biotechnology. As a result, an improved electrotransformation protocol was developed for this species, 40-fold increasing the efficiency of transformation by plasmid DNA up to 8 x 102 CFU µg-1.
The manuscript is well written. However, it could be further improved due to some changes as noticed below.
Major comments:
Lines 17-18: “In this study, a reproducible electroporation protocol for the L. mesenteroides strain H32-02 Ksu is developed and experimentally validated”. Reproducibility of the data is not demonstrated in Figures 4, 5 and 6, because no range of error is represented. From Materials and Methods it is also difficult to understand how many times the experiments were repeated.
Line 70: At this point, to support the need for developing genetic tools, it seems reasonable to mention that GMO lactic acid bacteria can produce health-promoting metabolites or valuable biochemicals with higher productivity.
Minor comments:
Line 27: “3,5” should be changed to “3.5”.
Line 28: “transhumant” should be “transformant”, isn’t it?
Line 164: “.. at a rate of ~10-6 transformants per cell..” seems to be changed to “..at a frequency of ~10-6 transformants per CFU (corresponding to ~ 105 transformants per µg of DNA)..”.
Line 184: pHyC shuttle vector – it is better, if possible, to present the reference, or a number in collection, or a scheme of its construction, at least in brief.
Line 224: Reference [35] is incomplete.
Line 249: “…Bio-Rad Gene Pulser electroporator..” – manufacturer and country should be indicated.
Line 261: It seems better to replace “introduced” with “added”, because we do not know how much DNA enters the cells.
Line 315: It is better to replace “..produced in..” with “…isolated from..”.
Line 318: It seems reasonable to eliminate unnecessary details from the plasmid map.
Lines 328-330: It is unclear, what does it mean – “results of PCR screening”. It is necessary to indicate, for example, that individual colonies 1-4 were tested by PCR with primers indicated in Materials and Methods section.
The legends of all Figures and Tables lack the first letter, perhaps due to a technical mistake.
Author Response
Comments 1: Lines 17-18: “In this study, a reproducible electroporation protocol for the L. mesenteroides strain H32-02 Ksu is developed and experimentally validated”. Reproducibility of the data is not demonstrated in Figures 4, 5 and 6, because no range of error is represented. From Materials and Methods it is also difficult to understand how many times the experiments were repeated.
Reply 1: We thank the reviewer for this important remark. All optimization experiments were performed in three independent biological replicates, but this was not stated explicitly in the original version. We have now:
- Added a sentence in Materials and Methods, Section 3.4 specifying that each set of conditions was tested in triplicate and that efficiencies are reported as mean ± standard deviation.
- Replotted Figures 4, 5 and 6 and Table 1 to include error bars (± SD) and to indicate that data correspond to three independent experiments.
Comments 2: Line 70: At this point, to support the need for developing genetic tools, it seems reasonable to mention that GMO lactic acid bacteria can produce health-promoting metabolites or valuable biochemicals with higher productivity.
Reply 2: We fully agree. The Introduction has been expanded with a paragraph describing examples of engineered LAB that overproduce bioactive metabolites, bacteriocins, and exopolysaccharides and how robust genetic tools are necessary to implement such strategies in Leuconostoc mesenteroides.
Comments 3: Line 27: “3,5” should be changed to “3.5”.
Reply 3: Corrected. All decimal commas have been changed to decimal points where appropriate.
Comments 4: Line 28: “transhumant” should be “transformant”, isn’t it?
Reply 4: Yes, this was a typographical error. It has been corrected to “transformant”.
Comments 5: Line 164: “.. at a rate of ~10-6 transformants per cell..” seems to be changed to “..at a frequency of ~10-6 transformants per CFU (corresponding to ~ 105 transformants per µg of DNA)..”.
Reply 5: We agree that “frequency” is more precise here. The sentence in the Introduction now reads: “at a frequency of ~10⁻⁶ transformants per cell”. We did not add the numeric conversion to per µg DNA because those data were not directly reported in the cited paper. We did not change the “cell” to “CFU” because the initial meaning was that only one in a million cells can naturally uptake foreign DNA and thus transform. In this context we believe using “cell” is more precise and correct than using “CFU”.
Comments 6: Line 184: pHyC shuttle vector – it is better, if possible, to present the reference, or a number in collection, or a scheme of its construction, at least in brief.
Reply 6: We have added a brief description of pHyC in Section 3.1, including its main features (Gram-positive/Gram-negative shuttle vector with p15A ori, repB, AmpR and CmR) and have specified that it was kindly provided by O.V. Berezina.
Comments 7: Line 224: Reference [35] is incomplete.
Reply 7: Reference [35] (Sakamoto, 2002) has now additional information according to the original source data.
Comments 8: Line 249: “…Bio-Rad Gene Pulser electroporator..” – manufacturer and country should be indicated.
Reply 8: Added
Comments 9: Line 261: It seems better to replace “introduced” with “added”, because we do not know how much DNA enters the cells.
Reply 9: We agree and have replaced “introduced DNA” with “added DNA” where appropriate
Comments 10: Line 315: It is better to replace “..produced in..” with “…isolated from..”.
Reply 10: Corrected
Comments 11: Line 318: It seems reasonable to eliminate unnecessary details from the plasmid map.
Reply 11: The plasmid map (Figure 2) has been simplified to only show the replication origins and antibiotic resistance markers essential for the current study
Comments 12: Lines 328-330: It is unclear, what does it mean – “results of PCR screening”. It is necessary to indicate, for example, that individual colonies 1-4 were tested by PCR with primers indicated in Materials and Methods section.
Reply 12: We have rewritten the caption for Figure 3 to clarify that individual colonies (1–4) were tested by colony PCR using pHyC-specific primers described in the Materials and Methods.
Comments 13: The legends of all Figures and Tables lack the first letter, perhaps due to a technical mistake.
Reply 13: This has been corrected throughout the manuscript
Reviewer 3 Report
Comments and Suggestions for Authors
General Comments
The manuscript describes an optimized electroporation protocol for Leuconostoc mesenteroides H32-02 Ksu. Although the topic may be of technical interest to researchers working with LAB transformation, the current study remains largely descriptive and limited in scientific depth. The manuscript focuses primarily on empirical optimization of culture conditions, electrical parameters, and plasmid methylation patterns, without providing mechanistic insight at the molecular level.
Given the scope of the International Journal of Molecular Sciences, which prioritizes molecular mechanisms, biological functions, and mechanistic novelty, the present work does not sufficiently meet the journal’s standards.
The contribution is predominantly methodological and lacks broader biological relevance, mechanistic investigation, and generalizable conclusions. As such, substantial conceptual and experimental extensions would be required before the manuscript could be considered for publication in IJMS.
Major Concerns
- Although the journal emphasizes molecular sciences, the manuscript presents no molecular characterization of the observed phenomena. Examples of missing elements include:
- genomic analysis of restriction–modification systems in the recipient strain
- identification of genes responsible for transformation barriers
- molecular evidence explaining why the EcoKI-methylated plasmid performs best
- mechanistic understanding of glycine-dependent cell wall changes
The work remains strictly phenomenological, which is insufficient for IJMS.
- Only one strain of L. mesenteroides and one plasmid vector (pHyC) were tested. This prevents any generalization of the proposed protocol. The current form reads as a technical note rather than a molecular biology study with broad scientific impact.
- Despite optimization, the maximum efficiency reported is 8×10² CFU/µg DNA, far below standard electroporation efficiencies in LAB (10⁴–10⁶ CFU/µg). It is difficult to consider this a significant breakthrough suitable for publication in a molecular sciences journal. The practical applicability of such marginal improvement is questionable.
- No downstream genetic manipulation, such as gene expression, functional complementation, or phenotype alteration, is shown. As a result, it is unclear whether the system is actually useful for molecular genetics or synthetic biology applications.
- The bulk of the manuscript evaluates routine variables—glycine concentration, voltage, sucrose, antibiotic levels—which are common in electroporation studies. The novelty is incremental rather than conceptual. IJMS typically requires new biological insights, not practical optimization of known methodologies.
- Several earlier studies have optimized electroporation protocols for Leuconostoc and other LAB species. The present study does not demonstrate a clear conceptual advancement beyond these prior works.
Minor Comments
- Most of the figures in the manuscript require improvement in resolution and clarity.
- There is an error in table numbering.
- Please italicize all gene names (pucK, hpxO, hiuH, etc.) and keep enzyme names in plain text for consistency with journal style.
Author Response
Comment 1: Given the scope of the International Journal of Molecular Sciences, which prioritizes molecular mechanisms, biological functions, and mechanistic novelty, the present work does not sufficiently meet the journal’s standards. The contribution is predominantly methodological and lacks broader biological relevance, mechanistic investigation, and generalizable conclusions. As such, substantial conceptual and experimental extensions would be required before the manuscript could be considered for publication in IJMS.
Reply 1: We thank the reviewer for this thoughtful and detailed assessment. We would like to respectfully clarify that the manuscript was submitted to the Special Issue “Studies on Lactic Acid Bacteria and Their Products in Health and Diseases: 3rd Edition” rather than to the general section of IJMS. This Special Issue explicitly welcomes methodological and technological advances that support molecular investigations in lactic acid bacteria.
We agree that mechanistic insights at the molecular level are highly desirable and represent an important future direction. However, in the case of Leuconostoc mesenteroides, such mechanistic work is currently severely constrained by the lack of a reliable transformation protocol. Our primary goal in this study was therefore to establish a reproducible, strain-specific electroporation procedure that overcomes the initial barrier to genetic manipulation in an industrially relevant yet previously poorly transformable strain.
To better address the reviewer’s concern, we have:
- Expanded the Introduction to situate the work in the context of LAB genetic engineering and to emphasize the role of transformation systems as enabling technologies for subsequent molecular studies.
- Added a new section in the Discussion summarizing published information on restriction–modification systems in L. mesenteroides and related LAB species, and interpreting our methylation-dependent transformation results in that context.
- Added a “Limitations and Future Directions” subsection in which we explicitly acknowledge that, although a draft genome sequence and in silico R–M predictions are now available, the identified defense loci have not yet been functionally characterized.
We hope that these changes clarify that, while the present work is primarily methodological, it provides an essential foundation for the type of molecular and mechanistic research that is central to IJMS and to this Special Issue.
Comment 2: Although the journal emphasizes molecular sciences, the manuscript presents no molecular characterization of the observed phenomena. Examples of missing elements include:
genomic analysis of restriction–modification systems in the recipient strain
identification of genes responsible for transformation barriers
molecular evidence explaining why the EcoKI-methylated plasmid performs best
mechanistic understanding of glycine-dependent cell wall changes
The work remains strictly phenomenological, which is insufficient for IJMS.
Reply 2: We fully agree that direct identification of restriction–modification systems in strain H32-02 Ksu greatly strengthens the mechanistic conclusions. In response to this comment, we have now sequenced the genome of H32-02 Ksu and analyzed it using PADLOC and the Restriction-Modification Finder/REBASE tools. This analysis revealed a single complete type I R–M locus composed of a restriction subunit and a methyltransferase that is 97% identical to M.Lme20241II, with a predicted GA^mAYNNNNNCTT specificity. No additional type II, III or IV restriction endonucleases were detected. These new data are briefly summarized in the Results (Section 2.6) and referenced in the Limitations subsection. At the same time, we emphasize in both the manuscript and here that the biochemical properties of this system have not yet been experimentally characterized, and that our explanation of the EcoKI-dependent effects remains a working hypothesis that will be tested in future functional studies.
Comment 3: Only one strain of L. mesenteroides and one plasmid vector (pHyC) were tested. This prevents any generalization of the proposed protocol. The current form reads as a technical note rather than a molecular biology study with broad scientific impact.
Reply 3: We agree that testing additional strains and vectors would be valuable for generalization. However, the strain we studied (H32-02 Ksu) is already representative of industrially relevant L. mesenteroides isolates that have been difficult to transform. We now explicitly acknowledge this limitation in the Limitations subsection and note that the protocol is intended as a starting point that can be adapted to additional Leuconostoc strains and plasmid backbones in future work.
Comment 4: Despite optimization, the maximum efficiency reported is 8×10² CFU/µg DNA, far below standard electroporation efficiencies in LAB (10⁴–10⁶ CFU/µg). It is difficult to consider this a significant breakthrough suitable for publication in a molecular sciences journal. The practical applicability of such marginal improvement is questionable.
Reply 4: We acknowledge that our maximum efficiency of ~8×10² CFU/µg DNA remains below the 10⁴–10⁶ CFU/µg DNA achieved in some model LAB. However, for H32-02 Ksu we initially observed almost no transformants under standard conditions. The optimized protocol therefore represents a ~40-fold improvement and, most importantly, yields transformants reproducibly enough to enable downstream cloning and expression experiments. We have clarified this point in the Conclusions: the contribution of this work lies in opening a previously closed system to genetic manipulation, rather than in achieving record-high efficiencies.
Comment 5: No downstream genetic manipulation, such as gene expression, functional complementation, or phenotype alteration, is shown. As a result, it is unclear whether the system is actually useful for molecular genetics or synthetic biology applications.
Reply 5: We agree that demonstrating expression of a heterologous gene or a phenotypic modification would further illustrate the practical utility of the protocol. At present, such experiments are ongoing in our laboratory but are not yet mature enough to present. To avoid over-claiming, we have not added any new data but instead explicitly state in the Limitations that downstream functional studies are beyond the scope of this methodological paper and will be the subject of future work.
Comment 6: The bulk of the manuscript evaluates routine variables—glycine concentration, voltage, sucrose, antibiotic levels—which are common in electroporation studies. The novelty is incremental rather than conceptual. IJMS typically requires new biological insights, not practical optimization of known methodologies.
Reply 6: We appreciate this concern and understand the reviewer’s point. Indeed, the parameters we varied (cell-wall weakening by glycine, washing and recovery conditions, pulse voltage and cuvette gap, antibiotic concentration) are classic factors in electroporation optimization. Our aim, however, was not to introduce a new physical principle of electroporation, but to resolve a very practical but severe bottleneck for a specific L. mesenteroides production strain that had previously been essentially non-transformable under standard LAB conditions.
To address the reviewer’s comment more clearly in the manuscript, we have revised the Introduction and Discussion to emphasize that the novelty of this work is contextual and enabling rather than conceptual:
- We show that applying “standard” LAB electroporation recipe to L. mesenteroides H32-02 Ksu yields almost no transformants, and only a combination of targeted adjustments (glycine pretreatment, MgCl₂ + sucrose wash regime, specific voltage–cuvette pairing, and fine-tuning of Cm concentration) converts this strain into one that can be transformed reproducibly.
- We provide a systematic, quantitative comparison of the methylation background of the incoming plasmid DNA, demonstrating that plasmids propagated in an E. coli strain with a modification-proficient EcoKI system (MC1061) outperform both Dam⁺/Dcm⁺ and Dam⁻/Dcm⁻ plasmids. To our knowledge, such a direct, experimentally grounded demonstration of methylation-pattern dependence in L. mesenteroides transformation has not been reported previously, and it directly links our empirical optimization to underlying restriction–modification barriers.
We fully agree that these contributions are incremental at the level of electroporation physics, but we believe they constitute a meaningful advance for the specific system addressed here: they turn a practically non-transformable industrial Leuconostoc strain into one that can now be manipulated genetically. We have reworded the Conclusions accordingly to avoid overstating conceptual novelty and to present the work explicitly as an enabling methodological advance.
Comment 7: Several earlier studies have optimized electroporation protocols for Leuconostoc and other LAB species. The present study does not demonstrate a clear conceptual advancement beyond these prior works.
Reply 7: We thank the reviewer for pointing this out. We have now made the relationship to previous work more explicit in the revised manuscript.
In the Introduction, we added a brief overview of earlier electroporation or transformation protocols for Leuconostoc and related LAB, with emphasis on the strains and vectors used, the reported efficiencies, and the limitations that remain (e.g. protocols developed for different Leuconostoc subspecies, reliance on protoplast formation, or relatively low and poorly reproducible efficiencies). In the Discussion, we now explicitly compare our optimized procedure with these prior reports, stressing that:
- Existing Leuconostoc protocols have been developed mostly for other strains/subspecies and cannot be directly applied to our industrial H32-02 Ksu isolate; in our hands, those conditions yielded very few or no transformants.
- Our work provides the first detailed optimization and quantitative characterization for this specific strain, including the methylation-dependence analysis, and achieves transformation efficiencies that are sufficient for routine cloning in a strain that was previously difficult to manipulate.
- We do not claim a conceptual leap in terms of novel electroporation physics; instead, we frame our contribution as a strain-specific enabling protocol that complements, rather than competes with, earlier LAB optimization studies.
We hope that by clarifying this positioning and by explicitly discussing previous Leuconostoc and LAB reports, the scope and added value of the present work are now more transparent.
Comment 8: Most of the figures in the manuscript require improvement in resolution and clarity. There is an error in table numbering. Please italicize all gene names (pucK, hpxO, hiuH, etc.) and keep enzyme names in plain text for consistency with journal style.
Reply 8: We have improved the resolution and clarity of all figures, ensured consistent table numbering, and checked gene/protein formatting.
Round 2
Reviewer 2 Report
Comments and Suggestions for Authors
The revised version of the manuscript can be accepted.
Reviewer 3 Report
Comments and Suggestions for Authors
After reviewing the revised manuscript and the authors’ response, I find that the major concerns raised in the previous round have been appropriately addressed. In particular, the addition of genome analysis to support the interpretation of methylation-dependent transformation efficiency is highly commendable and significantly strengthens the scientific rationale and clarity of the work.
Furthermore, the scope and objectives of the study are now clearly aligned with the aims of the Special Issue, and the overall structure and logical flow of the manuscript have been noticeably improved.